# A mixed-model approach for estimating drivers of microbiota community composition and differential taxonomic abundance

Amy R. Sweeny,[1,2] Hannah Lemon,[1] Anan Ibrahim,[3] Kathryn A. Watt,[1] Kenneth Wilson,[4] Dylan Z. Childs,[2] Daniel H. Nussey,[1] Andrew Free,[3] Luke McNally[1]

**ABSTRACT** Next-generation sequencing (NGS) and metabarcoding approaches are increasingly applied to wild animal populations, but there is a disconnect between the widely applied generalized linear mixed model (GLMM) approaches commonly used to study phenotypic variation and the statistical toolkit from community ecology typically applied to metabarcoding data. Here, we describe the suitability of a novel GLMM-based approach for analyzing the taxon-specific sequence read counts derived from standard metabarcoding data. This approach allows decomposition of the contribution of different drivers to variation in community composition (e.g., age, season, individual) via interaction terms in the model random-effects structure. We provide guidance to implementing this approach and show how these models can identify how responsible specific taxonomic groups are for the effects attributed to different drivers. We applied this approach to two cross-sectional data sets from the Soay sheep population of St. Kilda. GLMMs showed agreement with dissimilarity-based approaches highlighting the substantial contribution of age and minimal contribution of season to microbiota community compositions, and simultaneously estimated the contribution of other technical and biological factors. We further used model predictions to show that age effects were principally due to increases in taxa of the phylum Bacteroidetes and declines in taxa of the phylum Firmicutes. This approach offers a powerful means for understanding the influence of drivers of community structure derived from metabarcoding data. We discuss how our approach could be readily adapted to allow researchers to estimate contributions of additional factors such as host or microbe phylogeny to answer emerging questions surrounding the ecological and evolutionary roles of within-host communities.

**IMPORTANCE** NGS and fecal metabarcoding methods have provided powerful opportunities to study the wild gut microbiome. A wealth of data is, therefore, amassing across wild systems, generating the need for analytical approaches that can appropriately investigate simultaneous factors at the host and environmental scale that determine the composition of these communities. Here, we describe a generalized linear mixed-effects model (GLMM) approach to analyze read count data from metabarcoding of the gut microbiota, allowing us to quantify the contributions of multiple host and environmental factors to within-host community structure. Our approach provides outputs that are familiar to a majority of field ecologists and can be run using any standard mixed-effects modeling packages. We illustrate this approach using two metabarcoding data sets from the Soay sheep population of St. Kilda investigating age and season effects as worked examples.

**KEYWORDS** microbiota, metabarcoding, 16S, amplicon sequence variants, generalized linear mixed-effects model, community composition, differential abundance, Bayesian estimation

Address correspondence to Amy R. Sweeny, amyr.sweeny@gmail.com.

Amy R. Sweeny and Hannah Lemon contributed equally to this article. Author order was determined by Amy R. Sweeny's leading the initial draft writing.

Andrew Free and Luke McNally are joint last authors.

The authors declare no conflict of interest.

See the funding table on p. 13.

The ecological dynamics of within-host communities of parasites and commensal microbes can have dramatic effects on host health and fitness (1, 2). One increasingly well-studied example of such a within-host community is the so-called gut microbiota: the often complex and diverse community of commensal bacteria resident in the gastrointestinal tracts of their animal hosts. As well as playing a crucial role in the digestion of food, studies from humans and model laboratory animals highlight the impacts of the gut microbiota on host behavior and metabolism, as well as endocrine and immune homeostasis (3–6). A growing number of studies within ecology and evolutionary biology investigates the dynamics of the gut microbiota of natural systems using a combination of fecal sampling and next-generation sequencing (NGS) metabarcoding approaches. Understanding the role of within-host communities in underpinning host phenotypic variation, as well as wider ecological and evolutionary dynamics, in the wild will require statistical approaches that allow us to robustly quantify the contribution of different environmental and host-related factors to such metabarcoding data. Generalized linear mixed models (GLMMs) are a well-established and widely used suite of statistical models within ecology and evolution which provide a flexible means for appropriately dealing with the complex data structures and relationships between predictors of interest (7). Although they have yet to be widely applied in this context, GLMMs have huge potential to help dissect and understand the drivers of within-host community dynamics, as revealed by metabarcoding data.

Standard methodologies for investigating hypotheses concerning gut microbiota dynamics in the wild typically include the collection of fecal samples from selected study subjects and the application of NGS techniques for metabarcoding of informative bacterial genes for taxonomic assignment of sequenced reads (8). Microbiota community analysis commonly relies on the transformation of operational taxonomic units or amplicon sequence variant (ASV) counts into relative proportions per sample or rarefaction such that a set library size is randomly subsampled from all samples (9–11). Hypothesis testing using transformed counts from 16S taxonomic assignments typically is focused on community-level differences in taxonomic diversity and composition between experimental groups or time points of interest. Statistical approaches to this end include estimation of alpha diversity (the number of distinguishable taxa within a sample), distance measures (e.g., Bray–Curtis dissimilarity) (12), and ordination with dimensionality reduction (e.g., principal coordinates analysis [PCoA]). Data transformations and hypothesis tests in these approaches have several limitations. Standardizations of data based on proportions ignore heteroscedasticity from different library sizes across samples, while those relying on rarefaction restrict data such that the reads considered per each sample are limited to the minimum number of reads across all samples (9). This in turn can significantly elevate rates of false positives or reduce performance in microbiome clustering approaches. In addition to statistical pitfalls, these traditional approaches for assessing community-level differences differ philosophically from GLMM-based approaches that partition complex sources of variance. Although traditional approaches have provided substantial insights into microbiota community composition, they fall short of the flexibility and power offered by GLMM-based approaches to dissect the manifold and complex contributors to variation in measured phenotypes in natural populations. There has, therefore, been a movement among community ecologists toward such GLMM-based methods (13). Here, we develop a GLMM-based approach to decompose the sources of variation in count data derived from metabarcoding approaches and discuss the advantages of this approach for the analysis of microbiota and other community data.

The application of mixed-effects models to microbiota data sets is not new. The "Hierarchical Modeling of Species Communities" approach uses latent variable modeling and random effects to model community compositions and has been previously used to examine urbanization effects on fungal environmental microbiota (14). Similar approaches developing Joint Species Distribution Modeling for microbiota data sets have also shown insights into microbiota composition in the wild (15). Our suggested

approach differs from these in several ways. First, these approaches focus on modeling correlations among microbial taxa using latent variables to model residual correlation; this adds substantial complexity to the modeling process. Our approach does not attempt to model these correlations, and it focuses on variance decomposition of the sort familiar to ecologists and evolutionary biologists working on wild systems. If correlations among microbial taxa are of primary interest, we would direct readers to these approaches. Second, our approach does not require the use of any particular modeling package or a high degree of proficiency in coding. There are two central ideas in our approach—using sample-level random effects in over-dispersed Poisson models to account for variability in library size and using random effects of microbial taxonomy to allow for effects of host and environment on microbiota composition—that can be implemented in almost any random-effects modeling software or packages with which the reader is familiar. Thus, at the expense of modeling residual correlation among species, our approach offers a familiar method to decompose sources of variance in the microbiota for field scientists. Below we outline the motivation for this approach and illustrate this via an application to two 16S metabarcoding data sets from a wild mammal.

## MATERIALS AND METHODS

### A GLMM approach

As gut microbes have such important effects on host physiology, behavior, and health, much research has sought to identify individual microbial taxa that are responsible for alterations of host phenotype and state. This has in many ways mirrored the goals of many genome-wide association study (GWAS) analyses, which have sought to identify particular genetic variants associated with phenotypes of interest, often with a goal of developing diagnostics or drug targets (16). However, just as GWAS analyses have shown us that most phenotypes are highly polygenic, being determined by a complex combination of genetic variants of small effects (17–19), the study of host-associated microbiomes has often failed to find single taxa associated with host states (20, 21). Instead many changes in the host state are associated with general shifts in microbiome composition, often termed dysbiosis when accompanied by negative health consequences (20, 22). Phenomena such as dysbiosis shift the level at which we look for associations with host phenotype from a small number of microbial taxa to the whole microbiota. In addition, the most pressing questions about host-associated microbiota in ecology and evolution are very general and focused on the entire microbial community (2). For example, what are the relative roles of host physiology and environment in shaping the microbiota? How heritable is the microbiota? How much does microbiota composition impact fitness? The shift in focus of these questions from individual taxa to complete community poses an important conceptual and statistical challenge.

As previously discussed, host-associated microbiota often constitute hundreds or thousands of different taxa. Whenever we need to estimate a large ensemble of related parameters, a common statistical approach is to treat them as random variables from some distribution (23). To understand how this approach applies to the microbiota, let us consider the concrete question of estimating how the composition of the gut microbiota might change with season in a wild mammal. In traditional approaches to analyzing microbiota data sets, it would be common to visualize an ordination of the data, distinguishing points by season. Then, one would perform a permutational analysis of variance (ANOVA) on a dissimilarity matrix to test if microbiota from different seasons are more dissimilar than those from the same season and go on to test for the differential abundance of individual taxa across seasons to identify taxa with a major role in these changes (24, 25 ). In this approach, estimates for how individual taxa differ by season are all independent of each other. Using a random-effects model would approach this in a fundamentally different manner, where the effect of season across microbial taxa is treated as drawn from a random distribution. This approach has the advantage that all

taxa inform the estimate of the mean and variance of the distribution that the effects across taxa come from. The estimates of parameters for individual taxa are then "shrunk" to this distribution. This "shrinkage" is known to improve the accuracy of parameter estimation as long as there are large numbers of groups for the random effects, which is generally true for most host-associated microbiota owing to their large number of taxa.

While fitting such random-effects models is known to improve parameter estimation owing to shrinkage, its biggest advantage is in allowing us to shift the questions we ask to the whole microbiota level, and partition complex and inter-related sources of variance. GLMM approaches have been used across other ecological and evolutionary contexts to estimate repeatability, relative levels of spatiotemporal variance (26), social and common environment effects (27, 28), as well as heritability and the role of host genetics (29). Answering such questions has proved hugely challenging in the microbiota field as most analyses rely on tools which are not multi-level, from which it is extremely difficult to decompose the relative contribution of simultaneous processes at the host and environmental scale. However, multi-level models have been shown to offer significant advantages over many other compositional methods in community ecology for species abundance data (30). Here, we develop and illustrate a method to appropriately structure random effects across microbial taxa within a community using a GLMM, and thus partition the sources of variance driving microbiota composition.

To see how such a model can be structured, let us again return to the example of estimating the effect of season on the gut microbiota of a wild mammal (Fig. 1). Consider a scenario with two samples taken per host from a sample of hosts in a population, one in winter and one in summer, and with samples appropriately sequenced and reads bioinformatically assigned to ASVs. This will yield data in the form of a count of reads belonging to each ASV (the focal taxonomic group) within each sample (Fig. 1A), with two samples per individual host, one from each season. We can directly analyze such count data by fitting an over-dispersed Poisson family GLMM with a log-link. The predicted values on the link scale are given according to the following model (Fig. 1B).

$$log(y_{h, asv, s}) = \beta_0 + \beta_1 s + u_h + u_{asv} + u_{h:s} + u_{asv:h} + u_{asv:s} + u_{asv:h:s}$$

where $y_{h, asv, s}$ is the read count, $\beta_0$ is a global intercept, and the remaining terms account for technical variation effects in read counts (abundance) as well as biological variation in taxonomic composition. Fixed and random terms dealing with technical variation are as follows: $\beta_1$ is the effect of season ($s$) on total read count, $u_h$ is a random effect describing some variations in total read count among individual hosts ($h$, where there are multiple samples per host), $u_{asv}$ is a random effect describing variation in the total read count of each ASV across samples and hosts, $u_{h:s}$ is a random effect accounting for library size by describing variation in mean read count in each sample (i.e., host by season), and $u_{asv:h:s}$ is an additional random effect accounting for row-level variation (over-dispersion). In this example, biological effects of interest are specified as follows: $u_{asv:h}$ is a random effect describing the abundance (read count) of an ASV in host $h$, $u_{asv:s}$ is a random effect describing how ASV abundances (read counts) differ between seasons. By apportioning the variance attributed to these different random effects, we can assess the relative contributions of these different factors to microbiota composition (Fig. 1B and C). For example, a high variance associated with $u_{h:s}$ would indicate a high degree of technical variation due to library size variation across samples, and high variance associated with $u_{asv}$ could indicate high variation in read counts across ASVs due to over-dispersion of total abundance between common and rare taxa (Fig. 1C). With regard to biological inference, the variance associated with $u_{asv:h}$ can be interpreted as indicative of individual "repeatability" of ASV community composition and $u_{asv:s}$ can be interpreted as reflecting variance associated with compositional shifts across seasons.

Continuing with the above example, we can further use random-effect estimates from model outputs to explore which specific taxa are driving differential abundances

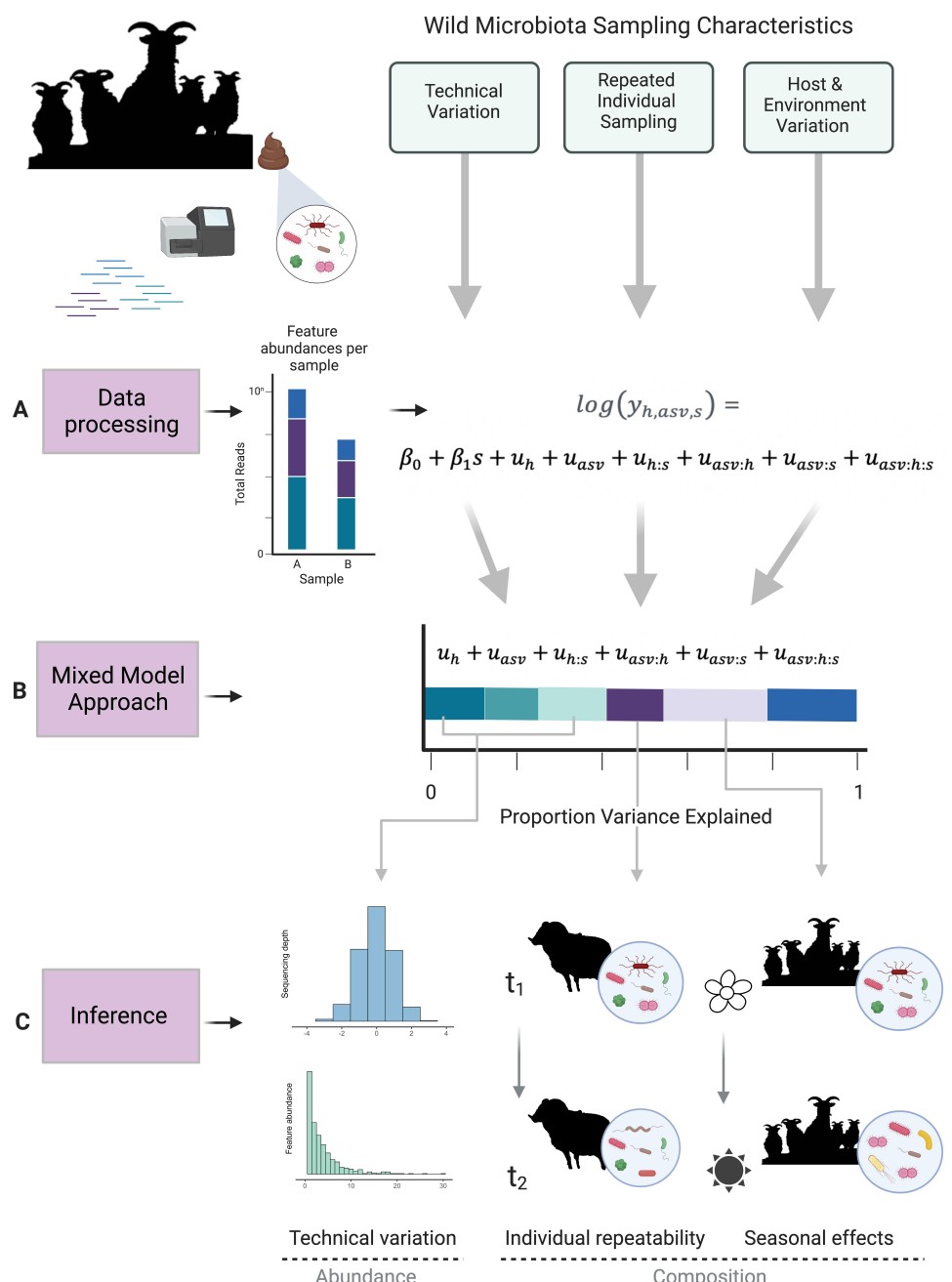

**FIG 1** Overview of mixed-model approach to wild microbiota analysis. Data processing (A) generates amplicon sequence variant (ASV)–level abundances for each sample. These raw abundances are used as the response for generalized linear mixed-effects models with Poisson error families. In the example illustrated, data include sampling time points for a group of individuals taken during two seasons. Model syntax therefore specifies a fixed effect of age, and random effects for taxonomy (asv), sample id (host:season, h:s), individual differential abundance of ASVs (asv:h), differential abundance of ASVs across seasons (asv:s), and a residual variance at the row level (asv:h:s). GLMM output can be used to partition the variance explained by each random-effect term (B). These variance components can be interpreted as the relative contributions of both technical variation and host or environmental contributions to differential abundance as illustrated in (C). Created with BioRender.com.

between groups of interests (e.g., season), which is commonly of great interest in microbiome studies, but for which many existing methods may be affected by library size and normalization methods (10, 31). Differential abundance in this example (and with Bayesian implementation) can be estimated using each ASV-by-season level of the random effect $u_{asv:s}$ and comparing posterior distributions for each ASV across factor

levels. Although fundamentally different from established significance tests of specific taxon abundances between groups of interest (e.g., references [25, 32, 33]), information from all taxa will influence the variance of random effects for holistic inference. For example, the mean of the posterior distribution for ASV1:summer minus that for ASV1:spring can be interpreted as the differential abundance of ASV1 between spring and summer, allowing identification of ASVs that exhibit the largest deviations from the means for further hypothesis generation and investigation.

While the Poisson model accounts for variation in library size across samples, there has also been a shift in microbiota research toward explicitly compositional data analysis, which removes any effects of library size (other than in quantifying uncertainty) prior to analysis. The centered log-ratio (CLR) described by Aitchison (34) represents one such transformation that may be useful for difficult data distributions or when complex random-effect structures are necessary. We present full details of how to implement the above GLMM approach using CLRs as an example of flexibility of this approach across error families and data transformations and then apply this alternative parameterization to the example data described below, in our supplementary files.

## A worked example: age and season effects on gut microbiota in wild sheep

To test and illustrate our approach, we obtained fecal samples from Soay sheep (*Ovis aries*) from the island of Hirta in the St. Kilda archipelago of the Outer Hebrides of Scotland. These animals are free-living and are part of a long-term study in which individuals have been marked and monitored longitudinally since 1985 (35). All animal sampled had been caught and uniquely tagged within a few days of birth so that their age and sex were known with certainty. Each year, fieldwork teams visit St. Kilda in spring to monitor lambing and capture, mark, and sample newborn lambs within a few days of birth. Subsequently, each August a larger field team visits to capture, mark, and sample animals living in the study area using a series of corral traps (35).

Two sets of fecal samples were collected, in 2013 and 2016, to allow comparison of the gut microbiota of individuals of different ages (2013) and from the same individuals sampled in different seasons (2016). The 2013 samples were collected during the August catch and included 30 samples from lambs (around 4 months old) and 28 samples from older adults (ages 2–13 years). The 2016 samples were collected from a set of 36 females aged 1–13 years who were sampled in both spring (around the time of parturition) and then 3–4 months later in August. Microbial DNA was extracted from samples, amplified using bacterial 16S rRNA V4 region primers, and sequenced using the Ilumina MiSeq platform to generate 250 base pair (bp) paired-end reads. Sequences were processed using the DADA2 pipeline in R (v1.12.1) to call ASVs (Callahan et al., 2016). Full details of sampling, sequencing, and data processing methods are provided in the electronic supplementary material (ESM 1.1 through 1.4). We conducted standard dissimilarity analysis and permutational multi-variate analysis of variance (PERMANOVA) tests on the effects of age and season for comparison to the results of our GLMM approach (see ESM 1.4.1 for full details). All data and code are available from GitHub (36).

## Specification of GLMMs

A tutorial describing the workflow for analysis can be found at https://arsweeny.github.io/microbiome-glmm/. We applied separate GLMMs to the 2013 and 2016 data sets. First, an aggregate data set for each year was created from the sample metadata, taxonomic classifications for each ASV, and an ASV-by-sample abundance matrix (*n* observations: 2016: 169,488; 2013: 117,102). We use a Bayesian framework and fit GLMMs with Poisson errors and log links to each data set using the package "MCMCglmm" (37) following the approach introduced in the section Materials and Methods. However, those wishing to use maximum likelihood estimation can do so using lme4 or ASREML (38, 39). Models fit to 2013 data, which included samples from hosts of two age classes from a single season (one sample per host) were specified as follows:

$$log(\bar{y}_{h,\,asv}) = \beta_0 + a\beta_a + u_h + u_{asv} + u_{asv:a} + u_{asv:h}$$

where $y_{h,\,asv}$ is the read count per ASV within each host, $\beta_0$ is a global intercept, $\beta_a$ is the effect of age ($a$, binary factor: lamb versus adult) on total read count, $u_h$ is a random effect describing variation in total read count among individual hosts ($h$, equivalent to sample here where hosts are sampled once each), $u_{asv}$ is a random effect describing variation in total read count of each ASV across hosts/samples, $u_{asv:a}$ is a random effect describing how ASV abundances (read counts) differ between host age classes, and $u_{asv:h}$ is an additional row-level random effect describing residual variation.

Models fit to 2016 data, which included samples from individual hosts of similar age sampled in both spring and summer of the same year (two samples per host), were specified as follows:

$$log(\bar{y}_{h,\,asv,\,s}) = \beta_0 + s\beta_s + u_h + u_{asv} + u_{h:s} + u_{asv:h} + u_{asv:s} + u_{asv:h:s}$$

Here $y_{h,\,asv,\,s}$ is the read count, $\beta_0$ is a global intercept, $\beta_s$ is the effect of season ($s$, binary factor: spring versus summer) on total read count, $u_h$ is a random effect describing variation in total read count among individual hosts ($h$, where there are multiple samples per host), $u_{asv}$ is a random effect by describing variation in total read count of each ASV across samples and hosts, $u_{h:s}$ is a random effect describing variation in total read count among individual hosts ($h$, where there are multiple samples per host), $u_{asv:h}$ is a random effect describing the abundance (read count) of an ASV in host $h$, $u_{asv:s}$ is a random effect describing how ASV abundances (read counts) differ between seasons, and $u_{asv:h:s}$ is an additional row-level random effect describing residual variation.

$\beta_0$ is a global intercept, and the remaining terms account for technical variation as well as biological variation of interest. Fixed and random terms dealing with technical variation are as follows: $\beta_1$ is the effect of season ($s$) on total read count, $u_h$ is a random effect describing variation in total read count among individual hosts ($h$, where there are multiple samples per host), $u_{asv}$ is a random effect by describing variation in the total read count of each ASV across samples and hosts, $u_{h:s}$ is a random effect accounting for library size by describing variation in mean read count in each sample (i.e., host by season), and $u_{asv:h:s}$ is an additional random effect accounting for row-level variation (over-dispersion). In this example, biological effects of interest are specified as follows: $u_{asv:h}$ is a random effect describing the abundance (read count) of an ASV in host $h$ and $u_{asv:s}$ is a random effect describing how ASV abundances (read counts) differ between seasons.

Using this GLMM approach, we calculated the relative contributions to sources of variance in the data from both technical and biological model components. We followed Nakagawa and Schielzeth (40) for the calculation of $r^2$ from GLMMs with Poisson error distributions. Using this formula, there is a portion of variance equal to 1 minus the sum variance of the model components, which represents variance arising from the Poisson distribution. Where multiple samples are present per individual (2016), repeatability of the community composition of ASVs can be estimated as the proportion of variance attributable to differential taxonomic composition across individuals divided by the sum of the variance explained by all other non-technical component terms estimating compositional effects $u_{asv:h}/(u_{asv:s} + u_{asv:h} + u_{asv:h:s})$.

We investigated differential abundances as outlined above. To extract information on specific bacterial taxa contributing to differential abundance across age groups or seasons, we used Poisson model outputs and subtracted the posterior distributions for each ASV between group levels (2013: age; 2016: season). We used the resultant distribution to calculate a mean difference and the highest posterior density interval (HPDI) to estimate differential abundance for each ASV. For example, the mean of the posterior distribution for ASV1:summer minus that for ASV1:spring can be interpreted as the differential abundance of ASV between spring and summer. A difference can be considered robust when credible intervals do not span zero.

## RESULTS

The gut microbiota communities of Soay sheep were dominated by two phyla, Firmicutes and Bacteroidetes (Fig. S1), as has been previously observed in most vertebrates (41). PCoA based on Bray–Curtis dissimilarity indicated clustering of samples by age and season (Fig. 2). The result of a PERMANOVA test on the 2013 data set showed a significant difference in group centroids for lambs and adults (pseudo-$F$ = 7.161, $P$ < 0.001), with 11.34% of the variance in gut microbiota composition ($R^2$) explained by differences between lambs and adults. PERMANOVA results for the 2016 data showed that group centroids for April and August are significantly distinct (pseudo-$F$ = 2.026, $P$ = 0.002), but season only explains 2.81% ($R^2$) of the observed variance.

Poisson GLMMs from 2013 data showed comparable results to ordination approaches, where community composition differed substantially between age classes (proportion variance $u_{asv:a}$: 19.88% CI 18.47%–21.39%; Fig. 3). Additional effects estimated by the model showed a substantial proportion of variance explained by taxonomic variation in ASV abundance ($u_{asv}$2013: 17.35%), a small portion of variation explained by variation in mean library size across samples ($u_h$ 2013: 1.59%) and considerable residual variance (estimated by the "units" term in MCMCglmm; $u_{asv:h}$ 2013: 49.42%; Fig. 3).

Poisson GLMMs from 2016 data likewise showed comparable results to ordination approaches, where community composition changed negligibly between seasons ($u_{asv:s}$: 1.24% CI 0.99%–1.44%; Fig. 3). The 2016 model showed a substantial proportion of variance explained by taxonomic variation in ASV abundance ($u_{asv}$ 2016: 34.74%), a small portion of variation explained by variation in mean library size across samples ($u_{h:s}$ 2016: 1.99%) and considerable residual variance ($u_{asv:h:s}$2016:46.67%;Fig.3).Repeated sampling of individuals in 2016 additionally showed moderate variance explained by inter-individual variation in community composition ($u_{asv:h}$ 2016: 5.49%). This equated to an individual repeatability of 11.1% for their microbiota community composition across sampling time points.

As outlined in Materials and Methods (Specification of GLMMs), we calculated differential abundances using the posterior distributions for each random-effect level of specific taxa across age classes (2013 data set; Fig. 4A and B) and seasons (2016 data set; Fig. 4C and D). For the ASV-by-age effect in the 2013 data ($u_{asv:a}$; Fig. 3), the estimates of taxa-specific differential abundances suggest that ASVs demonstrating strong shifts between lambs and adults belong primarily to two phyla, Firmicutes and Bacteroidetes (Fig. 4A and B). 683 out of 2,023 (33.76%) ASVs present in 2013 data showed shifts

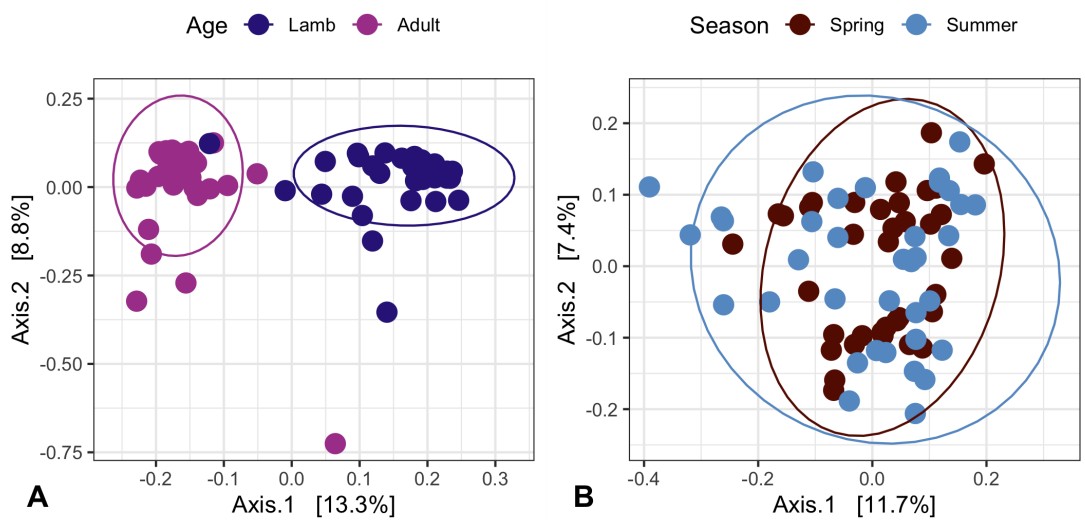

**FIG 2** Soay sheep gut microbiota beta diversity in adults and lambs from 2013 (A) and from April and August of 2016 (B). Principal coordinates analysis (PCoA) plots represent Bray–Curtis dissimilarity indicating clustering of samples by the group. Ellipsoids represent a 95% confidence interval surrounding each group.

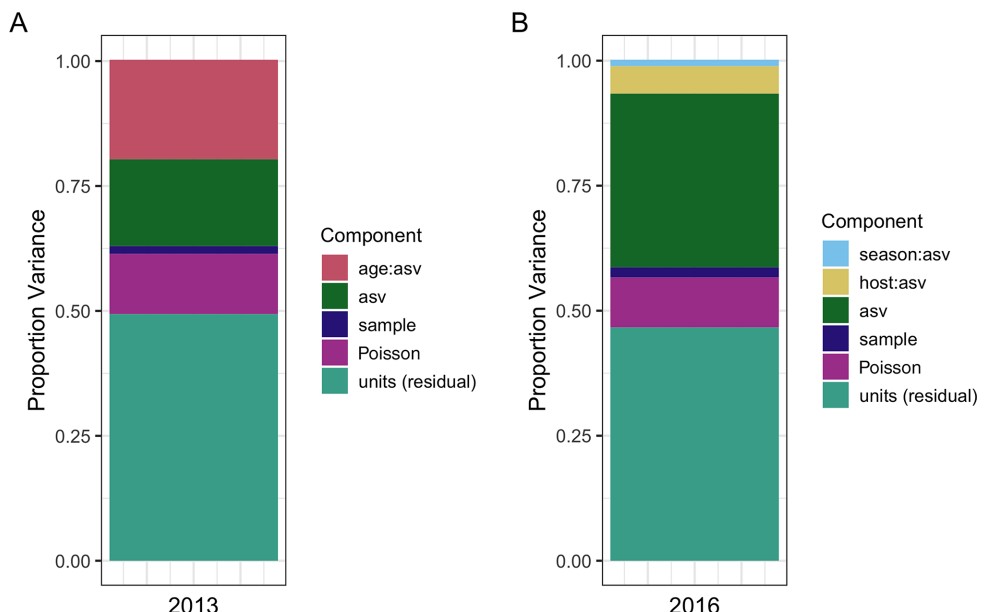

**FIG 3** Proportion of variance in bacterial read counts from different ASVs explained by GLMM component terms for two data sets. The 2013 data set (A) compared gut microbiota across two age classes from individuals sampled once at the same time point), while the 2016 data set (B) compared samples taken from the same individuals over two seasons.

between lambs and adults whose credible intervals did not span zero (50.81% positive shifts, 49.19% negative shifts). Bacteroidetes represented 52.16% of these positive shifts into adulthood, and Firmicutes represented 72.02% of these negative shifts into adulthood (Fig. 4B; Table S4). For the ASV-by-season effect in the 2016 data ($u_{asv:s}$; Fig. 3; Table S4), very few ASVs (24 of 2,364; 1.02%) had differential abundance effects with credible intervals that did not span zero between spring and summer sampling (Fig. 4C and D; Table S4).

Our results indicate that there are developmental shifts in the Soay sheep gut microbiota between lambs and adults and that the majority of taxa shifting in abundance belong to the Bacteroidetes and Firmicutes. However, this raises the question of whether this shift is because Bacteroidetes are generally more abundant in adults and Firmicutes more abundant in lambs, or if the ASVs that show these patterns just happen to be in these phyla. To illustrate how GLMMs can be used to address questions of this sort, we modified our models for the 2013 data set to include additional taxonomic effects of family and phylum, allowing us to identify taxonomic levels most responsible for differential abundances. Details of these phylogenetically more explicit GLMMs and their results and implications are presented in detail in ESM 1.4.3; Fig. S4 and Table S3. Results suggest considerable variation with respect to age across families and ASVs within both the Bacteroidetes and Firmicutes phyla and that most of the age effects on microbiota community composition occur at these lower taxonomic levels (Fig. S4; Table S3).

## DISCUSSION

We have described a novel approach to analyze metabarcoding data derived from NGS using a GLMM framework, have illustrated this method using data describing variation in the gut microbiota community in wild sheep, and have provided a user guide for implementing multiple versions of this approach (https://arsweeny.github.io/micro-biome-glmm/). Our approach represents an important step forward for researchers interested in using meta-taxonomic approaches to understand variation in the community structure in complex, non-experimental settings. It allows the well-established power and flexibility of GLMM-based approaches to be harnessed to decompose drivers of

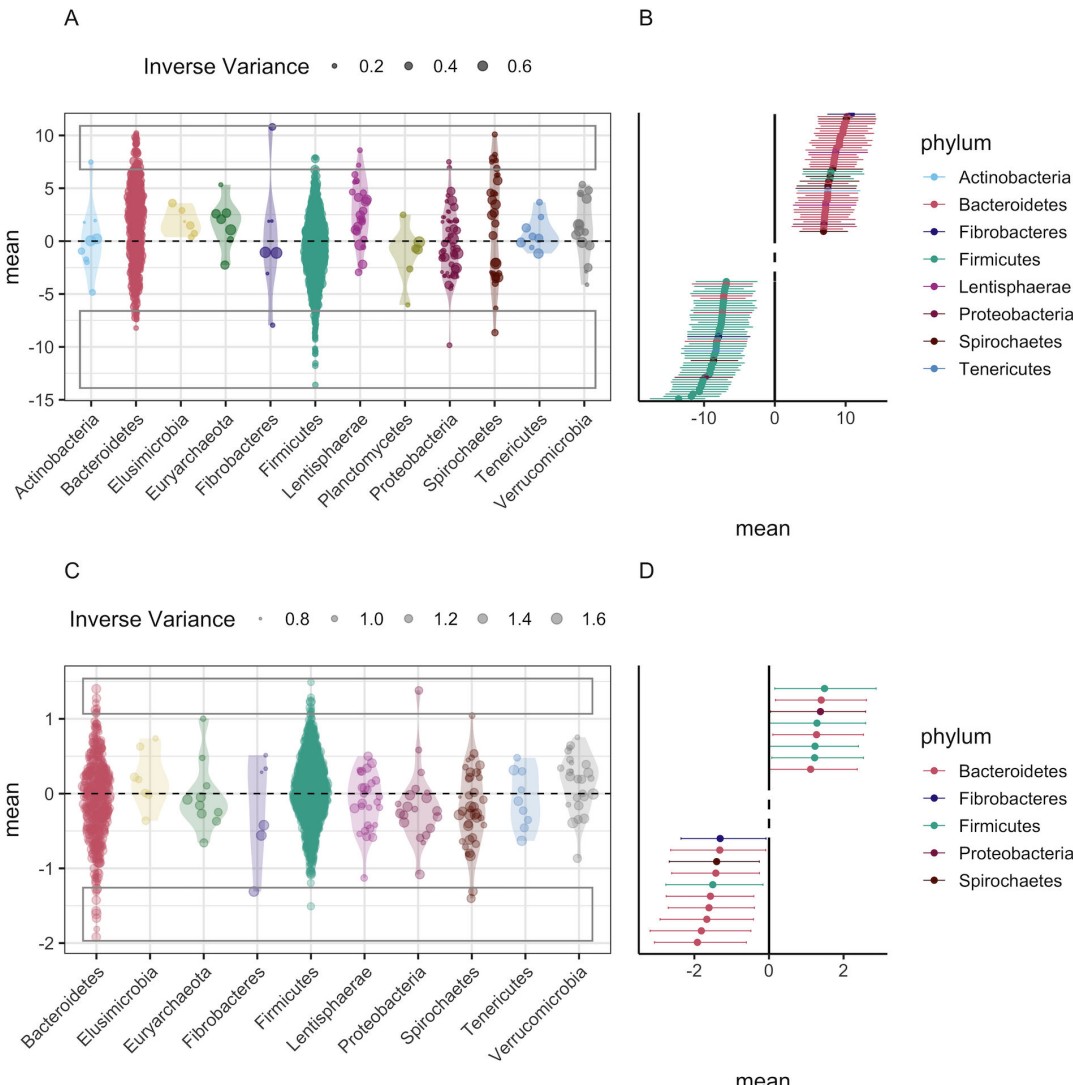

**FIG 4** Differential abundances across age classes (A and B) or season (C and D) for individual ASVs calculated from GLMMs with Poisson error families and taxonomic levels specified as ASV only. (A and C) represent all ASV-level effects. Violin plots represent the distribution of effect estimates, and size of the point represents the inverse variance of the estimate. Rectangles indicate the ASVs with the highest magnitude (positive or negative) differential abundances in forest plots (B and D). Forest plots represent point estimates and HPDI for the ASVs involved in the 50 (age class) or 10 (season) strongest increases and decreases of abundance.

variation in NGS-derived data on the taxonomic composition of samples. Our example analyses and tutorial provide very simple illustrations of how the approach can be used to estimate the contribution of host-related or environmental factors (specifically, age and season) to variation in the community structure of the gut microbiota. We show that results are comparable to those derived from widely applied ordination-based approaches and discuss the implications of observed variation in gut microbiota structure with age and season briefly below. However, these analyses are intended mainly as templates to help illustrate the approach, and barely scratch the surface of the types of important outstanding questions the method could be used to tackle with larger-scale data sets. Applying GLMMs to taxonomic-level sequence count data provides a rich toolkit from fields like quantitative genetics to dissect the contributions of different environmental and host factors to variation in community structure, with the potential to advance our understanding of community ecology, host–microbe or –pathogen interactions, and evolutionary dynamics.

In our illustrative analyses of wild Soay sheep, both GLMM and ordination-based approaches identified a stronger effect of age than a season on the gut microbiota. Changes in the structure of the gut microbiota across development in early life and during senescence in later adulthood are well-established in human studies (42) but remain poorly understood in natural populations. Our data clearly show the gut microbiota community structure changes between recently weaned lambs and adults and argues for further longitudinal studies in natural systems to test whether shifts in gut community structure could play a role in patterns of demographic aging in the wild. A growing number of studies in the wild have documented seasonal gut microbiota changes (43–46). The absence of strong seasonal differences in our data may be related to the relative homogeneity of the herbivorous diet of Soay sheep, as most previous wild studies are of omnivores with strong seasonal shifts in diet preference. Alternatively, it may be due to relatively low sample sizes in this pilot data set or because the spring and summer seasons we sampled are both periods of relatively high food abundance and quality compared to autumn and winter when habitat quality and food availability change more dramatically. In future studies, repeated sampling of the same individuals over time will be crucial to understand the effects of age, environmental, and other variables on gut community structure. The application of our approach to such longitudinal metataxonomic data sets will help researchers to robustly estimate within-individual patterns of change in community structures over time or space while also estimating how repeatable community structures across hosts.

Our novel GLMM approach allows the estimation of key ecological and evolutionary parameters from metabarcoding data sets which can advance our understanding of host–microbe evolutionary dynamics. Individual repeatability of measured phenotypes is an important and widely estimated parameter in ecology and quantitative genetics (47). Estimating the within-host repeatability of microbiota community structure over time can offer insight into the extent to which host control and environmental selection determine species composition (48). The GLMM structure presented here directly estimates this repeatability across two seasons in our 2016 data set at around 11%, although the small sample size and temporal proximity of the samples should mean we interpret this parameter estimate with caution. However, the model is illustrative and it should be clear that it is readily extendible to address emerging questions in the field. For example, the effects of host relatedness and inbreeding effects on microbiota composition have been explored previously in microbiome studies but via ordination methods using a small number of genetic clusters as grouping units (49). Including host genetic relatedness matrices as a random effect in a mixed-effects model (the so-called "animal models") within our GLMM framework can offer insights into heritability and inbreeding effects on community composition as it compares to other forces in the population (50). Factors that can be of both considerable ecological and evolutionary interest and that can also confound heritability estimates, such as maternal effects or shared nest or litter effects can likewise be incorporated into these models (51). Statistical advances to address maternal and social effects as well as spatial autocorrelation in ecological data sets can also be incorporated into microbiota analyses (52, 53). Applied to larger longitudinal data sets, our GLMM approach can allow researchers to directly estimate how different aspects of host state, genotype, and environment influence the structure of within-host communities and address many outstanding questions about the evolutionary and ecological causes and consequences of host–microbe interactions.

Full realization of the role of microbiota communities in the ecology and evolution of wild organisms depends on both identifying factors with important effects on global microbiota composition and on being able to test whether and how individual taxa or taxonomic groups underpin those effects. Our GLMM approach readily lends itself to addressing both questions. We have illustrated how the approach can be used to identify ASVs involved in community-level shifts with age identified in the random-effects structure of the 2013 models (Fig. 4) and to further decompose the contribution of

different taxonomic levels to community-level effects (ESM 1.4.3; Table S3). For example, our analysis highlights that analysis at the phylum level could provide a misleading view of compositional shifts associated with the age of Soay sheep and that there is substantial variation at the family level within each phylum (Fig. S4). This approach should offer similar insights to linear discriminant analysis (LDA) used in approaches such as LefSE (24), with the advantage of extraction of this information for multiple factors of interest rather than the requirement of a priori knowledge of effects of interest to run differential abundance analysis. Because variances are often unequal across taxonomic classes and nested taxonomic levels assume equal phylogenetic distance, our approach could be further developed to identify taxonomic levels associated with the greatest variance (ESM 1.4.3) by explicitly including the microbiota phylogeny within the GLMMs. This would allow the environment and host effects on community composition to be estimated accounting more accurately for phylogenetic distances between ASVs (54) in a similar manner to UNIFRAC clustering approaches (55). A GLMM-based approach capable of simultaneously dissecting the contributions of host environment, state, and genetics alongside microbial phylogeny to variation in microbial community structure seems to us to represent a powerful step toward to robustly address emerging questions surrounding the role of the microbiome in ecology and evolution.

Despite its advantages, we note that there are challenges with this approach and assumptions that must be considered in its application. In this paper, we employ several means of specifying parameters for and assessing the performance of models. In addition to the careful specification of random-effect structure aligned with the nature of the predictors, we encourage users employing MCMCglmm to inspect traces for model terms to identify autocorrelation or poor mixing to identify issues with convergence. In either Bayesian or frequentist frameworks, an inspection of model residuals can also indicate whether there are performance problems. Over-dispersion caused by zero inflation or aggregated counts can commonly pose problems for GLMMs (56), and over-dispersion is common in read count data describing microbiome community abundance (41). The degree of zero inflation and aggregation of counts will vary by site and system, and some consideration should be given to error family and data processing for model performance. In this manuscript, we use an over-dispersed Poisson distribution with an observation-level random effect; however, this does not always capture over-dispersion and can inflate $R^2$ values (57). Here, we used data subsets using several abundance thresholds in Poisson models to test the sensitivity of results to these choices (ESM 1.4.2). Additionally, we calculated the ratio of observed to model predicted zeros for both data sets presented in the main text and find that this ratio is close to 1 and that models predict true abundance means with very little deviation (ESM 1.4.4; Fig. S5). For instances where over-dispersed models may not suit investigators' data or where the inclusion of more complex random-effect structures introduces computational limitations, we also provide some discussion in the supplementary methods with a worked example of an alternate approach (ESM Section 2) which uses the CLR data transformation and Gaussian error families. Where zero inflation is notably high or researchers are interested in questions around both the abundance and prevalence of taxa within the microbiome, zero-inflated Poisson models are an additional option, although they can be difficult to fit and fall outside the scope of this introduction. As with any GLMM approach, there will be limitations to this method dependent on sample replication and distribution of data across levels of random effects (58). We also note that most implementations of mixed-effects models assume that the random effects come from a Gaussian distribution. While this may at first appear a strong assumption, GLMMs are generally quite robust to violations of this assumption, though there is an upward bias in variance estimates if the true distribution of effects is bimodal but modeled as Gaussian (59). Such problems should be identifiable from plotting data across levels of random effects of interest.

Beyond microbiota community analyses, approaches outlined in this manuscript are applicable more broadly to different types of metataxonomic data being

collected across myriad systems and research disciplines. For example, there has been great interest in describing the dynamics of the parasite community as an ecosystem and understanding its influence on host health (60–62). A growing number of studies apply metabarcoding to fecal samples to estimate the community structure of gastrointestinal parasite communities (63, 64), and our GLMM approach could readily be applied to such data sets to dissect the drivers of variation in parasite community composition. Another area of interest within ecology and evolution is using metabarcoding of fecal samples to estimate diet composition and its relationship to host phenotypes. Bayesian mixed-model approaches have also recently been applied to the analysis of the presence and absence of *Cyanistes caeruleus* (blue tit) diet components and align conceptually with approaches presented in this article (65). GLMM approaches to metabarcoding data maintain key similarities to other multivariate community ecological approaches to abundance data (66) while integrating the benefits of ecological and evolutionary approaches to quantifying phenotypic variation. We therefore suggest that approaches presented in this article can be applied across a range of systems and data types for powerful and flexible understanding of complex drivers of community dynamics.

## ACKNOWLEDGMENTS

This work was funded by a large Natural Environment Research Council (NERC) grant (NE/R016801/1), and the long-term study on St. Kilda was funded principally by responsive mode grants from NERC. L.M. was suppported by HFSP Young Investigator Project Grant RGY0072/21.

We thank Adam Hayward and Jill Pilkington for sample collection, the National Trust for Scotland for support of our work on St. Kilda, and QinetiQ and Kilda Cruises for logistical support. We also thank the Ecology Within Team for input in the analysis and manuscript and Josephine Pemberton for support and management of the field project. Figure 1 was created with Biorender.com. Photos on which sheep icons in Fig. 1 are based are by Hannah Vallin and Martin Stoffel.

L.M. and A.R.S. conceived and developed the statistical methods. A.R.S., L.M., A.F., and H.L. conducted the analyses. A.F., A.I., K.A.W., K.W., and D.H.N. oversaw and undertook sample collection and laboratory work. A.R.S. and H. L. wrote the first draft of the manuscript, and all authors contributed to the writing of the final manuscript.

## AUTHOR AFFILIATIONS

[1]Institute of Ecology & Evolution, University of Edinburgh, Edinburgh, United Kingdom
[2]School of Biosciences, University of Sheffield, Sheffield, United Kingdom
[3]Biochemistry and Biotechnology, Institute of Quantitative Biology, University of Edinburgh, Edinburgh, United Kingdom
[4]Lancaster Environment Centre, Lancaster University, Lancaster, United Kingdom

## AUTHOR ORCIDs

Amy R. Sweeny ⬤ http://orcid.org/0000-0003-4230-171X
Andrew Free ⬤ http://orcid.org/0000-0002-1787-4965
Luke McNally ⬤ http://orcid.org/0000-0002-1928-6235

## FUNDING

| Funder | Grant(s) | Author(s) |
| --- | --- | --- |
| UKRI \| Natural Environment Research Council (NERC) | NE/R016801/1 | Dylan Z. Childs |
| | | Daniel H. Nussey |
| | | Andrew Free |
| | | Luke McNally |

| Funder | Grant(s) | Author(s) |
|---|---|---|
| HFSP Young Investigator Project Grant | RGY0072/21 | Luke McNally |

## AUTHOR CONTRIBUTIONS

Amy R. Sweeny, Conceptualization, Data curation, Formal analysis, Investigation, Methodology, Visualization, Writing – original draft, Writing – review and editing | Hannah Lemon, Formal analysis, Investigation, Methodology, Writing – original draft, Writing – review and editing | Anan Ibrahim, Data curation, Investigation, Methodology, Writing – review and editing | Dylan Z. Childs, Conceptualization, Funding acquisition, Writing – review and editing | Daniel H. Nussey, Conceptualization, Funding acquisition, Investigation, Project administration, Supervision, Writing – original draft, Writing – review and editing | Andrew Free, Conceptualization, Data curation, Formal analysis, Funding acquisition, Investigation, Methodology, Validation, Writing – review and editing | Luke McNally, Conceptualization, Formal analysis, Funding acquisition, Investigation, Methodology, Supervision, Validation, Writing – original draft, Writing – review and editing.

## DATA AVAILABILITY

Data and code for this manuscript are available at GitHub. Sequences and metadata on which analysis is based can be found under number PRJEB39322 on the European Nucleotide Archive.

## ADDITIONAL FILES

The following material is available online.

### Supplemental Material

**Figure S1 (mSystems00040 S0001.tif).** Relative abundance of phyla and families in example data sets.
**Figure S2 (mSystems00040 S0002.tif).** Proportion variance across multiple levels of initial abundance filtering.
**Figure S3 (mSystems00040 S0003.tif).** Proportion variance for CLR models.
**Figure S4 (mSystems00040 S0004.tif).** Differential abundances across age classes accounting for multiple levels of taxonomy.
**Figure S5 (mSystems00040 S0005.tif).** Excess zero distributions.
**Table S1 (mSystems00040 S0006.xlsx).** MCMCglmm output for Poisson GLMMs.
**Table S2 (mSystems00040 S0007.xlsx).** MCMCglmm output for CLR Gaussian GLMMs.
**Table S3 (mSystems00040 S0008.xlsx).** MCMCglmm output for Poissonwith nested taxonomic structure.
**Table S4 (mSystems00040 S0009.xlsx).** Representation across phyla of significant ASV-level compositional shifts.
**Electronic Supplemental Information (mSystems00040 S0010.docx).** Additional information regarding methods, model specification, model validation, and alternate model implementations. Supplemental figure and table legends.

### Open Peer Review

**PEER REVIEW HISTORY (review-history.pdf).** An accounting of the reviewer comments and feedback.

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
