## [Reviewer comments · mSystems]

A mixed model approach for estimating drivers of microbiota community composition and differential taxonomic abundance

Amy Sweeny, Hannah Lemon, Anan Ibrahim, Kathryn Watt, Kenneth Wilson, Dylan Childs, Daniel Nussey, Andrew Free, and Luke McNally

Corresponding Author(s): Amy Sweeny, University of Sheffield

Review Timeline:

Submission Date:	February 8, 2023
Editorial Decision:	March 3, 2023
Revision Received:	May 4, 2023
Accepted:	May 8, 2023

Editor: Sarah Hird

Reviewer(s): The reviewers have opted to remain anonymous.

Transaction Report:

DOI: <https://doi.org/10.1128/msystems.00040-23>

Hi Amy -

The manuscript looks great. I marked it as "minor modifications" because I think you should (1) define what the triangles vs circles are in figure 2 and (2) check all figures for compatibility with the most common forms of colorblindness, especially when color is representing data. Figure 4 is especially hard to discern the colors for the different phyla. I hope these are easy "fixes" - the reviewer and I agree this is a great paper. Thank you for submitting to mSystems. (Form letter with additional details below.) Have a great weekend -
Sarah

March 3, 2023

Dr. Amy R Sweeny
University of Sheffield
School of Biosciences
Sheffield
United Kingdom

Re: mSystems00040-23 (A mixed model approach for estimating drivers of microbiota community composition and differential taxonomic abundance)

Dear Dr. Amy R Sweeny:

Thank you for submitting your manuscript to mSystems. We have completed our review and I am pleased to inform you that, in principle, we expect to accept it for publication in mSystems. However, acceptance will not be final until you have adequately addressed the reviewer comments.

Preparing Revision Guidelines

Sincerely,

Sarah Hird

Editor, mSystems

Journals Department
Reviewer comments:

Reviewer #1 (Comments for the Author):

The authors have made careful and considered revisions to the manuscript in line with my previous comments. The addition of a step-by-step tutorial will be very helpful to the field. I think this is an interesting and important paper that will be useful for microbiome researchers across a broad range of disciplines.

To the editors of mSystems,

Thank you for your time in reviewing our revised manuscript. We are delighted that the revisions made for an improved manuscript suitable for mSystems. We have made final requested changes to improve clarity of all figures. First, we removed shape aesthetic of Figure 2 as points are differentiated by colour. Second, we revised each figure to a safer palette for colour vision deficiency with care to choose distinct colours for categories in proximity where possible. We hope that with these changes, our manuscript is now suitable for publication.

Sincerely,

Amy R. Sweeny, PhD

May 8, 2023

Dr. Amy R Sweeny
University of Sheffield
School of Biosciences
Sheffield
United Kingdom

Re: mSystems00040-23R1 (A mixed model approach for estimating drivers of microbiota community composition and differential taxonomic abundance)

Dear Dr. Amy R Sweeny:

Your manuscript has been accepted, and I am forwarding it to the ASM Journals Department for publication. For your reference, ASM Journals' address is given below. Before it can be scheduled for publication, your manuscript will be checked by the mSystems production staff to make sure that all elements meet the technical requirements for publication. They will contact you if anything needs to be revised before copyediting and production can begin. Otherwise, you will be notified when your proofs are ready to be viewed.

If you would like to submit a potential Featured Image, please email a file and a short legend to msystems@asmusa.org. Please note that we can only consider images that (i) the authors created or own and (ii) have not been previously published. By submitting, you agree that the image can be used under the same terms as the published article. File requirements: square dimensions (4" x 4"), 300 dpi resolution, RGB colorspace, TIF file format.

We recognize that the video files can become quite large, and so to avoid quality loss ASM suggests sending the video file via <https://www.wetransfer.com/>. When you have a final version of the video and the still ready to share, please send it to mSystems staff at msystems@asmusa.org.

Sincerely,

Sarah Hird
Editor, mSystems

Journals Department
E-mail: mSystems@asmusa.org